# Performance Analysis of Feature Selection Methods in Software Defect Prediction: A Search Method Approach

**Abdullateef Oluwagbemiga Balogun [1,2,\*], Shuib Basri [1], Said Jadid Abdulkadir [1] and Ahmad Sobri Hashim [1]**

[1]   Department of Computer and Information Sciences, Universiti Teknologi PETRONAS, Perak 32610, Malaysia
[2]   Department of Computer Science, University of Ilorin, Ilorin 240103, Nigeria
\*   Correspondence: abdullateef_16005851@utp.edu.my or balogun.ao1@unilorin.edu.ng

**Abstract:** Software Defect Prediction (SDP) models are built using software metrics derived from software systems. The quality of SDP models depends largely on the quality of software metrics (dataset) used to build the SDP models. High dimensionality is one of the data quality problems that affect the performance of SDP models. Feature selection (FS) is a proven method for addressing the dimensionality problem. However, the choice of FS method for SDP is still a problem, as most of the empirical studies on FS methods for SDP produce contradictory and inconsistent quality outcomes. Those FS methods behave differently due to different underlining computational characteristics. This could be due to the choices of search methods used in FS because the impact of FS depends on the choice of search method. It is hence imperative to comparatively analyze the FS methods performance based on different search methods in SDP. In this paper, four filter feature ranking (FFR) and fourteen filter feature subset selection (FSS) methods were evaluated using four different classifiers over five software defect datasets obtained from the National Aeronautics and Space Administration (NASA) repository. The experimental analysis showed that the application of FS improves the predictive performance of classifiers and the performance of FS methods can vary across datasets and classifiers. In the FFR methods, Information Gain demonstrated the greatest improvements in the performance of the prediction models. In FSS methods, Consistency Feature Subset Selection based on Best First Search had the best influence on the prediction models. However, prediction models based on FFR proved to be more stable than those based on FSS methods. Hence, we conclude that FS methods improve the performance of SDP models, and that there is no single best FS method, as their performance varied according to datasets and the choice of the prediction model. However, we recommend the use of FFR methods as the prediction models based on FFR are more stable in terms of predictive performance.

**Keywords:** software defect prediction; feature selection; high dimensionality; search methods

## 1. Introduction

Software Defect Prediction (SDP) models are built using software metrics which based on data collected from the previous developed system or similar software projects [1]. Using such a model, the defect-proneness of the software modules under development can be predicted. The goal of SDP is to achieve high software quality and reliability with the effective use of available limited resources. In other words, SDP involves identifying software modules or components that are prone to defects. This will avail software engineers to prioritize the utilization of inhibited resources during each phase of the software development [2,3]. Consequently, reliability and quality for the software assessment, in

addition to software quality assurance, are guaranteed [4,5]. Software metrics which includes software source code complexity and development history are typically used to analyze the efficiency of the software process, quality, and reliability of the software products. In addition, software engineers use these software metrics for risk assessment and they are used for defect prediction to identify and improve the quality of software products [6–8]. Specifically, McCabe and Halstead Metrics, Procedural Metrics, Process Metrics, etc. are types of engineered software metrics used to determine the quality and reliability level of a software system [6,9]. A software module or component unit contains a set of features (metrics) and a class label. The class label depicts the state of the software module, either as defective or non-defective, while the derived features are used to develop SDP models [10,11]. That is, SDP uses historical data extracted from software repositories to determine the quality and reliability of the software modules or components [12,13].

SDP can be regarded as a classification task that involves categorizing software modules either as defective or non-defective, based on historical data and software metrics or features [14–16]. Software features or metrics reflect the characteristics of software modules. However, the numbers of metrics generated are usually high. This is due to various different types of software metric mechanisms used to determine the quality and reliability of a software system. The proliferation of these mechanisms consequently generates a large number of metric values, leading to a high-dimensionality problem when many feature values are generated. In addition, some of these features (metrics) may be more relevant to the class (defective or non-defective) than others, and some may be redundant or irrelevant [17,18].

Feature selection (FS) can be used to select high uncorrelated features from the high dimensional features. In other words, it can select those features that are more relevant and irredundant to the class label of the dataset among the features. Therefore, introducing FS methods into SDP can solve the high dimensionality problem [17–19]. FS is a vital data pre-processing step in classification processes as it improves the quality of data and consequently improves the predictive performance of the prediction models. Existing research has shown that irrelevant features, along with redundant features, can severely affect the accuracy of the defect predictors [20–23]. Thus, there is a need for finding an efficient feature selection method which can identify and remove as much irrelevant and redundant information as possible, thereby leading to good predictive performance with low computational cost [24,25]. Supervised feature selection techniques evaluate the available feature's characteristics and derive a set of pertinent characteristics based on labeled datasets. The criteria used to determine the useful feature characteristics depend on the underlining computational characteristics of the technique utilized. Filter feature-ranking (FFR) methods which are types of supervised FS methods depend on the computational characteristics of each feature by certain critical factors, and then the analyst culls some features that are congruous with a particular dataset. On the other hand, filter feature subset selection (FSS) methods (another type of supervised FS methods) search for a subset of features that have good predictive capability collectively. In this study, a comparative performance analysis of FFR and FSS methods based on different search methods are investigated to determine their respective efficacy in culling a germane set of features.

Recent studies have compared the impact of FS methods on the performance of SDP [26–32]. Some studies conclude that some of the FS methods are better than others [27,28,30,31], while some studies claimed that there is no significant difference between the performances of FS methods in SDP [26,29,32]. This contradiction and inconsistency in results by existing studies may be due to the choice of search mechanism used in FS methods.

In this study, four different FFR methods (Information Gain (IG), ReliefF (RFA), Gain Ratio (GR), and Clustering Variation (CV)) based on Ranker Search method and two FFS methods: Correlation-based Feature Subset Selection (CFS) and Consistency Feature Subset Selection (CNS) based on two different search approaches: Exhaustive Search (Best First Search (BFS) and Greedy Stepwise Search (GSS) methods) and Heuristic Search (Genetic Search (GS), Bat Search (BAT), Ant Search (AS), Fire-Fly Search (FS), and Particle Swarm Optimization (PSO) method). Four classification techniques: Naïve

Bayes (NB), Decision Tree (DT), Logistic Regression (LR), and K-Nearest Neighbor (KNN) were used to evaluate the effectiveness of these FS methods. The respective models were used on five software defects dataset from the NASA repository and their predictive performances were measured comparatively based on accuracy.

From our experimental results, the application of FS improves the predictive performance of the prediction models and the performance of FS methods varies across datasets and prediction models. IG recorded the best improvement on prediction models over other FFR methods, while CNS based on BFS had the best influence on prediction models based on FSS methods. Further analysis showed that prediction models based on FFR are more stable in terms of performance accuracy than other FS methods.

The rest of this paper is structured as follows. Section 2 presents a literature review and analysis of existing related works. Section 3 presents the various FS methods including the search methods, classifiers, datasets, and the performance metric considered in the experimental works of this study. Section 4 highlights the experimental procedure, experimental results, and discussion of our findings for the experimental works. Section 5 presents the threats to the validity of this study. Section 6 concludes the comparative study and summarizes future work.

## 2. Related Works

A major problem associated with SDP is the dilemma of having a large number of metrics (features). In other words, using all software metrics in training an SDP model can end up negatively affecting the predictive performance of the model. As such, many FS approaches have been proposed in addressing the selection of optimal software metrics. Some studies went to the extent on comparing these methods in order to identify the best method. However, most of these studies yielded contradictory and inconsistent conclusions on the effect of FS methods in SDP [26,29,32].

Ghotra et al. [28] performed a large scale impact analysis of twenty-eight FS methods on twenty-one commonly used classifiers. Their experiment was based on software defect datasets from the NASA and the PROMISE repositories. They concluded that correlation-based filter-FS method based on the BF search method outperforms other FS methods across the datasets. This is a good indicator that their experiment covered a large number of FS methods, classification techniques, and datasets. However, they only considered BF and GA search methods as search mechanisms for the FSS methods. There are other heuristics and meta-heuristic search methods such as BAT, AS, FS, etc. that may perform better than BF and GS in this context.

Afzal and Torkar [26] conducted a benchmark study by empirically comparing state-of-the-art FS methods. They have considered, IG, RF, Principal Component Analysis (PCA), CFS, CNS and wrapper subset evaluation (WRP). NB and DT were deployed on Five software defect datasets and the predictive models were evaluated based on Area Under Curve (AUC). Their results showed that FS is beneficial to SDP but there was no individual best FS method for SDP. This could be partly due to the number and type of software defect datasets been considered and the choice of search mechanisms in the case of the FSS and WRP FS methods.

Gao et al. [27], regarded feature selection as a search problem in software engineering. Their study was concerned with software quality estimation and they proposed a hybrid FS approach which was based on Kolmogorov–Smirnov statistic and automatic hybrid search (AHS). Their results showed that AHS was superior to other methods, and that an elimination of 85% of software metrics may affect performances of SDP models positively or remain constant.

Akintola et al. [18] performed a comparative analysis of classifiers based on FFS on SDP and their results gave credit to the usage of FFS, but there can still be further analysis using other FS methods. It has been proven empirically that wrappers obtain subsets with better performance than filter feature selection because the subsets were evaluated using a real modeling algorithm [33,34]. Rodriguez et al. [35] have also conducted comparative experiments on FS methods based on three

different FFR and WRP models on four software defect datasets. Their results showed that smaller data sets generally maintain predictability with fewer features than the original data sets.

From the aforementioned studies, only a handful of comparative performance studies have been carried out to evaluate the efficacy of FS methods based on different search mechanisms. Therefore, there is a need to have a vital comparative evaluation of FS methods based on different search mechanisms in SDP. This is to create a better understanding of FS methods characteristics and guide researchers and analysts on the selection of search methods in FS based for SDP. In this paper, a comparative performance analysis of eighteen FS methods in SDP is presented. Each FS methods was used with four different classifiers selected based on performance and heterogeneity. The respective SDP models were tested with five software defect datasets from NASA repository and evaluated based on prediction accuracy. The performance stability of each prediction model based on FS methods was further evaluated via the coefficient of variation for each prediction models.

## 3. Methodology

This section describes the FS methods, the respective search methods, classification algorithms, experimental setup, software defect dataset and performance metrics used in this study.

### 3.1. Filter Feature Ranking Method

Filter Feature Ranking (FFR) method uses the computational characteristics of datasets to independently assess and rank attributes in datasets which are found to be independent of the prediction model. It grades each attributes base on different characteristics such as statistics, probability, instance or classifier based indicators. Attributes are thereafter selected based on their score [29]. In this paper, four FFR methods were considered based on different functional characteristics. For the selection of top-ranked features, $\log_2 N$ was used in this study, where $N$ is the number of software metrics in the full software defect dataset. Table 1 presents a description of the FFR methods used in this study.

**Table 1.** List of Filter Feature Ranking (FFR) Methods.

| Filter Feature Rank Method | Search Method | Characteristics | Reference |
|---|---|---|---|
| Information Gain Attribute Evaluator (IG) | Ranker Search | Probability-based | [18,32] |
| Relief Feature Attribute Evaluator (RFA) | Ranker Search | Instance-based | [29,32] |
| Gain Ratio Attribute Evaluator (GR) | Ranker Search | Probability-based | [29,32] |
| Clustering Variation Attribute Evaluator (CV) | Ranker Search | Statistics-based | [18,29] |

### 3.2. Filter Feature Subset Selection Method

Feature Subset Selection (FSS), just like the FFR, assesses, ranks and select features based on some properties. However, in the case of FSS, the major focus is the search method which is used to generate a subset of features that collectively have good prediction potentials. It considers the existence of better predictive performance when a feature is combined with other features other [32]. Two FSS methods were considered in this study. As mentioned earlier, FSS is based on various search methods. Therefore, these search methods traverse the feature space to generate a subset with high predictive potentials. Consequently, the performance of FSS varies per the search methods [29]. Table 2 presents a detailed description of the FSS methods used in this study, and Table 3 shows the various search methods with their parameters used in the FSS methods.

**Table 2.** List of Filter Feature Subset Selection (FSS) Methods.

| Filter Feature Subset Selection Method | Search Method | Reference |
|---|---|---|
| Correlation-based Feature Subset Selection (CFS) | Best First Search (BFS) | [32,36] |
| | Greedy Stepwise Search (GSS) | [29,36] |
| | Ant Search (AS) | |
| | Bat Search (BAT) | |
| | Firefly Search (FS) | |
| | Genetic Search (GS) | [32,36] |
| | PSO Search (PSOS) | |
| Consistency Feature Subset Selection (CNS) | Best First Search (BFS) | [32,36] |
| | Greedy Stepwise Search (GSS) | [29,36] |
| | Ant Search (AS) | |
| | Bat Search (BAT) | |
| | Firefly Search (FS) | |
| | Genetic Search (GS) | [32,36] |
| | PSO Search (PSOS) | |

**Table 3.** Various Search Methods and Parameter Setting.

| Search Methods | Parameter Settings |
|---|---|
| Best First Search | Direction = Bi-directional |
| Greedy Stepwise Search | Conservative Forward Selection = True; Search Backwards = False; NumToSelect = $\log_2 N$ (N = Number of Features) |
| Ant Search | AccelerateType = Accelerate; Chaotic co-efficient = 0.4 Chaotic Parameter Type = Chaotic Map: Parameter; Chaotic Type = logistic map; PopulationSize = 200; Phromone = 2.0 |
| Bat Search | AccelerateType = Accelerate; Chaotic co-efficient = 0.4; Chaotic Parameter Type = logistic Map: Parameter; Chaotic Type = logistic map; PopulationSize = 200; loudness = 0.5 |
| Firefly Search | AccelerateType = Accelerate; Chaotic co-efficient = 0.4; Chaotic Parameter Type = logistic Map: Parameter; Chaotic Type = logistic map; PopulationSize = 200; absorption = 0.001; betaMin = 0.33 |
| Genetic Search | PopulationSize = 200; MaxGeneration = 20; crossoverProb = 0.6 |
| PSO Search | PopulationSize = 200; IndividualWeight = 0.34; InertiaWeight = 0.33; SocialWeight = 0.33 |

*3.3. Classification Algorithms*

For the classification process, four widely used classification algorithms were considered for evaluating the efficacy of the FS methods. These are Naïve Bayes (NB), Decision Tree (DT), Kernel Logistic Regression (LR), and K Nearest Neighbor (KNN). This is in line with the aim of this study to evaluate the efficacy of FFR and FSS methods, which should be independent of the classification technique. The four classifiers were selected based on different characteristics: NB based on Bayes' theorem, DT from Tree-based methods, LR from the function-based classification technique and KNN from the instance based learning classifiers. The heterogeneity of the selected classifiers is to investigate how different FS methods perform on different classifiers with different characteristics. Table 4 gives a brief description of these classification algorithms with respect to their classification characteristics and their parameter settings.

**Table 4.** Classification Algorithms.

| Classifiers | Description | Parameter Setting |
|---|---|---|
| Naïve Bayes (NB) | A Bayes Theorem based classification technique. | NumDecimalPlaces = 2; NumAttrEval = Normal Dist. |
| Decision Tree (DT) | A Tree-based classification technique. | Confidence factor = 0.25; MinNumObj = 2 |
| Kernel Logistic Regression (LR) | A Function-based classification technique. | Kernel = PolyKernel (E = 1.0; C = 250007); lambda = 0.01; Quadratic penalty = BFGS Optimization |
| K Nearest Neighbor (KNN) | An Instance learning-based classification technique. | K = 1; NNSearch = LinearNNSearch (based on Euclidean Distance) |

### 3.4. Experimental Setup

This section discusses the experimental setup of the comparative performance analysis of FS methods as depicted in Figure 1. The experimental setup can be described based on three major steps.

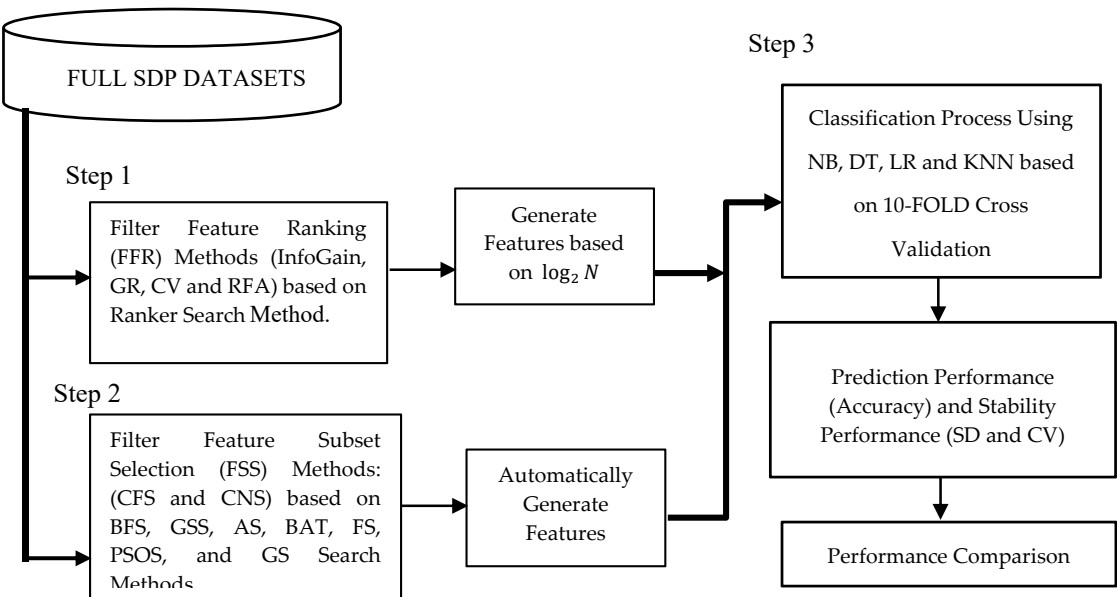

**Figure 1.** Experimental Setup.

**Step 1**: At first, four FFR methods as presented in Table 1 are applied to the original full software defect datasets. In this case, each of the FFR methods (IG, RFA, GR, and CV based on Ranker Search method) assesses and ranks the software features based on their respective characteristics. $\log_2 N$ features were selected from the ranked list provided by each of the FFR methods. Specifically, the first six features were selected from the resulting rank list of the FFR methods. The $\log_2 N$ (where N is number of features in each software defect datasets) was adopted according to the work of Gao et al. [27] which indicated that it is better to cull $\log_2 N$ features; many empirical studies have followed this procedure [26,28,29,32]. According to Table 4, the value for *N* in this study is (*N* >= 21) and this is due to the choice of different software defect datasets for the experiment. In the end, reduced datasets in terms of attributes are generated.

**Step 2:** Secondly, the two FSS methods in Table 2 and the respective search methods in Table 3 are applied to the original full datasets. The two FSS (CFS, CNS) methods with seven search methods (BFS, BAT, FS, AS, GSS, PSOS, and GS) making fourteen FSS methods to apply on the dataset. Each of

the FSS methods generates a subset of features that have high predictive potentials. The respective parameter settings of the search methods as depicted in Table 3 were used. As in the case of FFR methods, reduced datasets are also generated at the end of this step.

**Step 3**: This step involves the prediction process. That is the application of classifiers in Table 4 on the software defect dataset (Filtered datasets from Step 1 and 2). Four classifiers (NB, DT, LR, and KNN) were applied on both the reduced (Filtered datasets from Step 1 and 2) and full datasets. The essence of this step is to show the efficacy of reduced software features in SDP. Each of the experiments was carried out using the 10-folds cross-validation (CV) method. This is to avoid the issue of biases and overfitting of the prediction models and also to reduce the effect of class imbalance which is one of the data quality problems in data mining [29,37]. In addition, due to the random nature of the search methods of the FSS methods, each experiment involving FSS methods were performed 10 times to validate the result. Eventually, a total of 2900 ((4 FFR methods × 5 datasets × 4 classifiers) + (14 FSS methods × 5 datasets × 4 classifiers × 10 runs) + (4 classifiers × 5 datasets)) distinct experiments were carried out in this study.

### 3.5. Software Defect Datasets

The datasets used in this study are obtained from the National Aeronautics Space Administration (NASA) Facility Metrics Data Program (MDP) repository [38]. Recently, studies have shown that these datasets are noisy and need to be cleaned and pre-processed [26,28,29,32]. The cleaned version of the NASA datasets from Shepperd et al. [39] was used in this study. Table 5 presents the description of the clean NASA datasets with their number of features and modules.

**Table 5.** Software Defect Datasets.

| S/No. | Datasets | Language | Number of Features | Number of Modules |
|:---:|:---:|:---:|:---:|:---:|
| 1. | CM1 | C | 37 | 327 |
| 2. | KC1 | C++ | 21 | 1162 |
| 3. | KC3 | Java | 39 | 194 |
| 4. | MW1 | C | 37 | 250 |
| 5. | PC2 | C | 37 | 722 |

### 3.6. Performance Evaluation Metrics

In this study, the performance evaluation method is based on accuracy which measures the percentage of the correctly classified instances. The metric values were computed using the statistical values of True Positive (TP), True Negative (TN), False Positive (FP), and False Negative (FN).

$$\text{Accuracy} = \frac{\text{TP} + \text{TN}}{\text{TP} + \text{FP} + \text{FN} + \text{TN}} \times 100\% \tag{1}$$

To determine the performance stability of prediction models, Co-efficient of Variation (C.V) was applied to the results of the prediction models. C.V which is the percentage ratio of standard deviation (SD) and average (AVE) is used to remove the effect of average difference on the comparison stability [15,40]. The formula for C.V is given as thus:

$$\text{C.V} = \frac{\text{SD}}{\text{AVE}} \times 100\% \tag{2}$$

Prediction models with high C.V values are regarded as unstable.

## 4. Experimental Results

This section presents the experimental results based on the experimental procedure (See Figure 1). The performances of each prediction models were analyzed based on accuracy and the results were compared on two cases (with and without application of FS methods). The parameter settings for each classifier and search method are as shown in Tables 3 and 4. All prediction models were built based on the WEKA machine learning tool [41]. For replication of this study, all models and datasets used in this study are readily available in [42].

As mentioned previously, four FFR and fourteen FSS methods were applied one after the other to five software defect datasets. Four classifiers based on the selected and full features of the datasets were used to develop the SDP models. Tables 6–9 present the accuracy values of the four classifiers on the software defect datasets. These accuracy values were from two scenarios (With FS methods and Without FS methods). Specifically, in each of the tables, the accuracy performance value of each classifier on the individual datasets (full and reduced) is shown. The search methods for the FSS (CFS and CNS) were further grouped into two (i.e. Heuristic method and Exhaustive method). This is to show the distinction of how the respective search methods behaved when used as subset evaluation methods in FSS. The average performance of these classifiers (NB, DT, LR, and KNN) across all datasets were computed and the variation of the average performance of each prediction models with FS (FFR and FSS) methods to the average performance of each prediction model without FS methods were also computed. This is to show how significant the effectiveness of the application of FS methods in SDP. As shown in Tables 6–9, it is observed that the accuracy performance of the prediction models based on FS methods, in this case, FFR and FSS were better than when no FS methods are applied. This further strengthens the evidence that FS methods can improve the performance of prediction models in SDP.

**Table 6.** Accuracy Values of Naive Bayes Classifier on Full and Reduced Datasets.

| Naïve Bayes (NB) | Models | Performance Metrics (Accuracy)/Dataset | | | | | Average (%) | Variation (%) |
|---|---|---|---|---|---|---|---|---|
| | | CM1 | KC1 | KC3 | MW1 | PC2 | | |
| NO Feature Selection | NB | 81.35 | 73.58 | 78.87 | 81.60 | 90.30 | 81.14 | 0 |
| NB + CFS + Heuristic Search | NB+GA | 84.10 | 73.84 | 80.41 | 84.4 | 94.18 | 83.39 | 2.77 |
| | NB+BAT | 85.32 | 74.96 | **81.96** | 86 | 95.01 | **84.65** | 4.33 |
| | NB+PSO | 83.18 | 73.84 | 79.90 | 84.8 | 94.46 | 83.24 | 2.58 |
| | NB+FS | 82.87 | 73.92 | 80.41 | 84.8 | 93.91 | 83.18 | 2.52 |
| | NB+AS | **86.49** | 75.13 | 80.41 | 85.2 | 95.57 | 84.56 | 4.22 |
| NB + CFS + Exhaustive Search | NB+GSS | 83.79 | 73.58 | 80.93 | 84.4 | 93.91 | 83.32 | 2.69 |
| | NB+BF | 83.18 | 73.84 | 79.90 | 84.8 | 94.32 | 83.21 | 2.55 |
| NB + CNS + Heuristic Search | NB+GA | 81.96 | 73.49 | 79.38 | 85.2 | 94.18 | 82.84 | 2.10 |
| | NB+BAT | 81.65 | 73.49 | 78.87 | 84 | 94.46 | 82.49 | 1.67 |
| | NB+PSO | 82.87 | 73.41 | 80.41 | 85.6 | 94.87 | 83.43 | 2.83 |
| | NB+FS | 81.35 | 73.41 | 80.41 | 83.6 | 93.63 | 82.48 | 1.65 |
| | NB+AS | 82.87 | 73.49 | 79.38 | 83.6 | 94.46 | 82.76 | 2.00 |
| NB + CNS + Exhaustive Search | NB+GSS | 81.65 | **75.39** | 79.90 | 86 | 93.91 | 83.37 | 2.75 |
| | NB+BF | 85.32 | 73.32 | 80.93 | 84.4 | **97.78** | 84.35 | 3.96 |
| NB + Filter Method | NB+IG | 84.10 | 74.78 | 80.93 | 83.6 | 93.91 | 83.46 | 2.87 |
| | NB+RFA | 80.12 | 72.63 | 78.87 | 84.4 | 96.68 | 82.54 | 1.73 |
| | NB+GR | 84.10 | 73.24 | 80.41 | 84.4 | 94.18 | 83.27 | 2.62 |
| | NB+CV | 81.96 | 74.61 | 79.38 | **86.4** | 95.84 | 83.64 | 3.08 |

Boldface typeface indicates the highest value for each dataset.

**Table 7.** Accuracy Values of Decision Tree Classifier on Full and Reduced Datasets.

| Decision Tree (DT) | Models | Performance Metrics (Accuracy)/Dataset | | | | | Average (%) | Variation (%) |
|---|---|---|---|---|---|---|---|---|
| | | CM1 | KC1 | KC3 | MW1 | PC2 | | |
| NO Feature Selection | DT | 81.04 | 74.18 | 79.38 | 90.40 | 97.51 | 84.50 | 0 |
| DT + CFS + Heuristic Search | DT+GA | 85.32 | 75.82 | **82.99** | 89.6 | 97.51 | 86.25 | 2.06 |
| | DT+BAT | 87.16 | 74.96 | **82.99** | 90.4 | **97.78** | **86.66** | **2.55** |
| | DT+PSO | 86.54 | 75.65 | 81.96 | 89.2 | 97.51 | 86.17 | 1.97 |
| | DT+FS | 85.93 | 75.22 | 80.41 | 88.4 | 97.65 | 85.52 | 1.21 |
| | DT+AS | 85.93 | 75.13 | 82.47 | 89.6 | 97.51 | 86.13 | 1.92 |
| DT + CFS + Exhaustive Search | DT+GSS | 86.54 | 74.10 | 80.41 | 89.2 | 97.51 | 85.55 | 1.24 |
| | DT+BF | 86.54 | 75.47 | 81.44 | 89.6 | 97.51 | 86.11 | 1.91 |
| DT + CNS + Heuristic Search | DT+GA | 86.24 | 73.75 | 80.93 | 89.2 | 97.65 | 85.55 | 1.24 |
| | DT+BAT | 85.63 | 75.30 | 80.41 | 89.2 | 97.51 | 85.61 | 1.31 |
| | DT+PSO | 86.24 | 74.44 | 80.93 | 88.4 | 97.65 | 85.53 | 1.22 |
| | DT+FS | 83.49 | 74.44 | 78.87 | 89.6 | 97.65 | 84.81 | 0.36 |
| | DT+AS | 85.93 | 74.44 | 80.93 | 89.2 | 97.37 | 85.57 | 1.27 |
| DT + CNS + Exhaustive Search | DT+GSS | 87.16 | 75.56 | 80.41 | **91.6** | 97.65 | 86.47 | 2.33 |
| | DT+BF | 86.85 | 74.44 | 80.93 | 89.6 | 97.78 | 85.92 | 1.68 |
| DT + Filter Method | DT+IG | 86.54 | 74.96 | 82.47 | 90 | 97.65 | 86.32 | 2.16 |
| | DT+RF | 86.24 | **76.25** | 79.90 | 87.6 | **97.78** | 85.55 | 1.24 |
| | DT+GR | 86.24 | 75.04 | 81.96 | 90.4 | **97.78** | 86.28 | 2.11 |
| | DT+CV | **87.46** | 74.01 | 78.87 | 88.8 | **97.78** | 85.38 | 1.04 |

Boldface typeface indicates the highest value for each dataset.

**Table 8.** Accuracy Values of Logistic Regression Classifier on Full and Reduced Datasets.

| Logistic Regression (LR) | Models | Performance Metrics (Accuracy)/Dataset | | | | | Average (%) | Variation (%) |
|---|---|---|---|---|---|---|---|---|
| | | CM1 | KC1 | KC3 | MW1 | PC2 | | |
| NO Feature Selection | LR | 85.32 | 76.76 | 82.47 | 88.00 | 97.09 | 85.93 | 0 |
| LR + CFS + Heuristic Search | LR+GA | 85.93 | 75.82 | 80.93 | 90 | 97.51 | 86.04 | 0.12 |
| | LR+BAT | 85.32 | 75.56 | 81.44 | **91.2** | 97.51 | 86.21 | 0.32 |
| | LR+PSO | **87.16** | 76.16 | 82.47 | 89.6 | 97.65 | 86.61 | 0.79 |
| | LR+FS | 86.24 | 75.47 | **82.99** | 90 | 97.51 | 86.44 | 0.60 |
| | LR+AS | 85.93 | 75.47 | 82.47 | 88.8 | **97.78** | 86.09 | 0.19 |
| LR + CFS + Exhaustive Search | LR+GSS | 86.24 | 76.76 | 82.47 | 90 | 97.51 | 86.60 | 0.78 |
| | LR+BF | 86.85 | 75.90 | 82.47 | 89.2 | 97.65 | 86.41 | 0.56 |
| LR + CNS + Heuristic Search | LR+GA | **87.16** | 76.59 | **82.99** | 90.8 | 97.37 | **86.98** | **1.22** |
| | LR+BAT | 85.02 | 76.08 | 81.44 | 89.6 | 97.51 | 85.93 | 0.00 |
| | LR+PSO | 85.02 | 76.51 | 81.44 | 90.4 | 97.65 | 86.20 | 0.32 |
| | LR+FS | 85.32 | 76.51 | 82.47 | 90 | 97.23 | 86.31 | 0.44 |
| | LR+AS | 84.71 | **77.11** | 81.96 | 90.4 | 97.23 | 86.28 | 0.41 |
| LR + CNS + Exhaustive Search | LR+GSS | 86.54 | 76.33 | 81.96 | 89.6 | 97.65 | 86.42 | 0.57 |
| | LR+BF | **87.16** | 76.51 | 81.96 | 90.4 | **97.78** | 86.76 | 0.97 |
| LR + Filter Method | LR+IG | 86.54 | 75.47 | 81.96 | 90.4 | 97.51 | 86.38 | 0.52 |
| | LR+RF | **87.16** | 76.42 | 80.93 | 89.2 | 97.65 | 86.27 | 0.40 |
| | LR+GR | **87.16** | 75.82 | 81.96 | 88.8 | 97.09 | 86.16 | 0.27 |
| | LR+CV | 86.85 | 75.65 | 80.41 | 90.4 | 97.51 | 86.16 | 0.27 |

Boldface typeface indicates the highest value for each dataset.

**Table 9.** Accuracy Values of K Nearest Neighbor Classifier on Full and Reduced Datasets.

| K Nearest Neighbor (KNN) | Models | Performance Metrics (Accuracy)/Dataset | | | | | Average (%) | Variation (%) |
|---|---|---|---|---|---|---|---|---|
| | | CM1 | KC1 | KC3 | MW1 | PC2 | | |
| **NO Feature Selection** | KNN | 77.98 | 73.24 | 72.16 | 83.60 | 95.71 | 80.54 | 0 |
| **KNN + CFS + Heuristic Search** | KNN+GA | 77.68 | 71.26 | **78.87** | 84 | 96.54 | 81.67 | 1.40 |
| | KNN+BAT | 77.37 | 72.98 | **78.87** | 84.4 | 96.12 | 81.95 | 1.75 |
| | KNN+PSO | 80.43 | 70.57 | 75.77 | 84 | 96.81 | 81.52 | 1.21 |
| | KNN+FS | 81.04 | 70.40 | 74.23 | 84 | 95.84 | 81.10 | 0.70 |
| | KNN+AS | 78.59 | 71.69 | 75.26 | 82 | 95.29 | 80.57 | 0.03 |
| **KNN + CFS + Exhaustive Search** | KNN+GSS | 78.90 | 71.34 | 77.32 | 84 | 96.40 | 81.59 | 1.31 |
| | KNN+BF | 80.43 | 70.40 | 75.77 | 82.8 | 96.81 | 81.24 | 0.87 |
| **KNN + CNS + Heuristic Search** | KNN+GA | 81.35 | 73.49 | 74.74 | 83.6 | 96.26 | 81.89 | 1.68 |
| | KNN+BAT | 79.20 | **73.75** | 77.84 | 84.4 | 96.12 | 82.26 | 2.14 |
| | KNN+PSO | 79.51 | 73.49 | 78.87 | 82.8 | 97.09 | 82.35 | 2.25 |
| | KNN+FS | 80.73 | 73.58 | 75.26 | 82.8 | 96.12 | 81.70 | 1.44 |
| | KNN+AS | 76.15 | 72.81 | 76.29 | 83.6 | 96.40 | 81.05 | 0.63 |
| **KNN + CNS + Exhaustive Search** | KNN+GSS | 77.68 | 69.88 | **78.87** | 85.6 | 95.84 | 81.57 | 1.28 |
| | KNN+BF | **85.32** | 73.67 | 77.32 | 84.4 | **97.78** | **83.70** | **3.92** |
| **KNN + Filter Method** | KNN+IG | 77.37 | 70.57 | 73.71 | 84.8 | 96.26 | 80.54 | 0.00 |
| | KNN+RF | 81.64 | 73.06 | 68.56 | 84 | 96.12 | 80.68 | 0.17 |
| | KNN+GR | 76.76 | 71.86 | 74.23 | 84.4 | 96.12 | 80.67 | 0.17 |
| | KNN+CV | 77.98 | 69.97 | 74.74 | **86.8** | 95.98 | 81.09 | 0.69 |

Boldface typeface indicates the highest value for each dataset.

Specifically, considering the average accuracy performance value of prediction models based on NB classifier as shown in Table 6, NB with CFS using BAT heuristic search method had the highest average accuracy value, i.e., 84.65%. This accuracy value is better than when no FS methods are used with the NB model (81.14%) by 4.33%. The same goes to prediction models based on DT classifier, as shown in Table 7. DT with CFS using BAT heuristic search method had the best average accuracy value of 86.66%. Compared with when no FS methods on DT (84.50%), a variation of 2.33% increment was observed. In the case of LR classifier, as presented in Table 8, LR with CNS based on GS had the highest average accuracy value of 86.98% with a positive variation of 1.22% when compared with when no FS methods are used on LR (85.93%). From Table 9, KNN with CNS based on BFS had the highest average accuracy value of 83.70% with a positive variation of 3.92% when compared with no FS methods. It was also observed that LR with CNS based on GS had the highest average accuracy value (85.93%) across all prediction models and NB with CFS based on BAT heuristic search had the highest positive variation (4.33%). This clearly shows that FS methods have a positive effect on the prediction models as the average accuracy values based on each classifier without FS methods are less than when FS methods are applied. Our findings on the positive effect of FS methods on prediction models are in accordance with research outcomes from existing empirical studies. Ghotra et al. [28], Afzal and Torkar [26], and Akintola et al. [18] in their respective studies also reported that FS methods had a positive effect on prediction models in SDP. However, our study explored the effect of FS methods on prediction models based on the search method which is different from existing empirical studies.

Furthermore, assessing the accuracy performance of each prediction model on each of the dataset will showcase how these prediction models perform based on different FS methods. Tables 10–14 present the comparisons of FS (FFR and FSS) methods on each of the five datasets respectively. Considering the number of features generated by FS methods, FFR features are pre-calculated based on $\log_2 N$ (where $N$ is the number of features). The number of features for FSS methods are based on the search methods used (Heuristic or Exhaustive). Across all datasets and FS methods used in this study, the number of features generated by CFS is less than CNS. On the CM1 dataset, with respect to CFS, LR with PSO search method and DT with BAT search method had an accuracy value of 87.16% with (LR+CFS+PSO) selecting eight features and the (DT+CFS+BAT) had five features. Same also was

observed for CNS, LR with GA search, DT with GSS search, and LR with BF search had accuracy value of 87.16%. LR with GA search had more features (twelve) and LR with BF selected just one feature. However, based on the FFR method, DT with CV based on Ranker search had the highest accuracy value of 87.46% which means the FFR method gave the best performance on CM1 dataset as presented in Table 10. From Table 11, LR with CNS based on AS (77.11%) out-performs all prediction models on the KC1 dataset. The prediction model was built on seventeen features as selected by AS. Other FS methods on KC1 selected smaller features but their respective prediction model had lower accuracy performance. In addition, on dataset KC3 as presented in Table 12, DT with CFS based on BAT and GA search had the highest accuracy value of 82.99% with two features selected while LR with CFS based on BAT search in Table 13 had the best accuracy value on MW1 dataset. In Table 14, DT and LR with CFS based on BAT and AS respectively had an accuracy value of 97.78% on PC2 dataset. The FFR methods also had similar accuracy value on PC2 based on DT with RFA, GR, and CV. Clearly, there was no significant difference in the performance of the FS methods, as their respective performance and effect varies from dataset to dataset and the choice of classification algorithm. This research outcome is related to the findings from Xu et al. [29], Kondo et al. [31] and Muthukumaran et al. [30]. Although on average, FSS methods proved to be better than FFR methods.

**Table 10.** Performance Accuracy Values of FS-based Prediction Models on CM1 dataset.

| | | | FILTER-BASED SUBSET SELECTION METHODS (FSS) | | | |
|---|---|---|---|---|---|---|
| | | | **CfsSubsetEval (CFS)Method** | | | |
| **Attribute Evaluator** | **Search Methods** | **No. of Features** | **Performance Metrics (Accuracy)/Classifier** | | | |
| | | | **NB** | **DT** | **LR** | **KNN** |
| **Heuristics Method** | GA | 7 | 84.10 | 85.32 | 85.93 | 77.68 |
| | BAT | 5 | 85.32 | **87.16** | 85.32 | 77.37 |
| | PSO | 8 | 83.18 | 86.54 | **87.16** | 80.43 |
| | FS | 7 | 82.87 | 85.93 | 86.24 | 81.04 |
| | AS | 5 | 86.49 | 85.93 | 85.93 | 78.59 |
| **Exhaustive Method** | GSS | 5 | 83.79 | 86.54 | 86.24 | 78.90 |
| | BF | 5 | 83.18 | 86.54 | 86.85 | 80.43 |
| | Average | | 84.13 | 86.28 | **86.24** | 79.20 |
| | | | **ConsistencySubsetEval (CNS)Method** | | | |
| **Attribute Evaluator** | **Search Methods** | **No. of Features** | **Performance Metrics (Accuracy)/Classifier** | | | |
| | | | **NB** | **DT** | **LR** | **KNN** |
| **Heuristics Method** | GA | 12 | 81.96 | 86.24 | **87.16** | 81.35 |
| | BAT | 12 | 81.65 | 85.63 | 85.02 | 79.20 |
| | PSO | 6 | 82.87 | 86.24 | 85.02 | 79.51 |
| | FS | 15 | 81.35 | 83.49 | 85.32 | 80.73 |
| | AS | 8 | 82.87 | 85.93 | 84.71 | 76.15 |
| **Exhaustive Method** | GSS | 6 | 81.65 | **87.16** | 86.54 | 77.68 |
| | BF | 1 | 85.32 | 86.85 | **87.16** | 85.32 |
| | Average | | 82.53 | **85.93** | 85.85 | 79.99 |
| | | | FILTER-BASED FEATURE RANKING METHODS (FFR) | | | |
| **Attribute Evaluator** | **Search Methods** | **No. of Features** | **Performance Metrics (Accuracy)/Classifier** | | | |
| | | | **NB** | **DT** | **LR** | **KNN** |
| **IG** | Ranker | 6 | 84.10 | 86.54 | 86.54 | 77.37 |
| **RFA** | Ranker | 6 | 80.12 | 86.24 | 87.16 | 81.64 |
| **GR** | Ranker | 6 | 84.10 | 86.24 | 87.16 | 76.76 |
| **CV** | Ranker | 6 | 81.96 | **87.46** | 86.85 | 77.98 |
| | Average | | 82.57 | 86.62 | **86.93** | 78.44 |

Boldface typeface indicates the highest value for each classifier.

**Table 11.** Performance Accuracy Values of FS-based Prediction Models on KC1 dataset.

| FILTER-BASED SUBSET SELECTION METHODS (FSS) | | | | | | |
|---|---|---|---|---|---|---|
| | CfsSubsetEval (CFS) Method | | | | | |
| Attribute Evaluator | Search Methods | No. of Features | Performance Metrics (Accuracy)/Classifier | | | |
| | | | NB | DT | LR | KNN |
| Heuristics Method | GA | 8 | 73.84 | 75.82 | 75.82 | 71.26 |
| | BAT | 4 | 74.96 | 74.96 | 75.56 | 72.98 |
| | PSO | 8 | 73.84 | 75.65 | 76.16 | 70.57 |
| | FS | 4 | 73.92 | 75.22 | 75.47 | 70.40 |
| | AS | 2 | 75.13 | 75.13 | 75.47 | 71.69 |
| Exhaustive Method | GSS | 6 | 73.58 | 74.10 | **76.76** | 71.34 |
| | BF | 8 | 73.84 | 75.47 | 75.90 | 70.40 |
| | Average | | 74.16 | 75.19 | **75.88** | 71.23 |
| | ConsistencySubsetEval (CNS) Method | | | | | |
| Attribute Evaluator | Search Methods | No. of Features | Performance Metrics (Accuracy)/Classifier | | | |
| | | | NB | DT | LR | KNN |
| Heuristics Method | GA | 11 | 73.49 | 73.75 | 76.59 | 73.49 |
| | BAT | 17 | 73.49 | 75.30 | 76.08 | 73.75 |
| | PSO | 16 | 73.41 | 74.44 | 76.51 | 73.49 |
| | FS | 16 | 73.41 | 74.44 | 76.51 | 73.58 |
| | AS | 17 | 73.49 | 74.44 | **77.11** | 72.81 |
| Exhaustive Method | GSS | 6 | 75.39 | 75.56 | 76.33 | 69.88 |
| | BF | 16 | 73.32 | 74.44 | 76.51 | 73.67 |
| | Average | | 73.72 | 74.63 | **76.52** | 72.95 |
| FILTER-BASED FEATURE RANKING METHODS (FFR) | | | | | | |
| Attribute Evaluator | Search Methods | No. of Features | Performance Metrics (Accuracy)/Classifier | | | |
| | | | NB | DT | LR | KNN |
| IG | Ranker | 6 | 74.78 | 74.96 | 75.47 | 70.57 |
| RFA | Ranker | 6 | 72.63 | 76.25 | **76.42** | 71.86 |
| GR | Ranker | 6 | 73.24 | 75.04 | 75.82 | 71.86 |
| CV | Ranker | 6 | 74.61 | 74.01 | 75.65 | 69.97 |
| | Average | | 73.82 | 75.06 | **75.84** | 71.06 |

Boldface typeface indicates the highest value for each classifier.

**Table 12.** Performance Accuracy Values of FS-based Prediction Models on KC3 dataset.

| FILTER-BASED SUBSET SELECTION METHODS (FSS) | | | | | | |
|---|---|---|---|---|---|---|
| | CfsSubsetEval (CFS)Method | | | | | |
| Attribute Evaluator | Search Methods | No. of Features | Performance Metrics (Accuracy)/Classifier | | | |
| | | | NB | DT | LR | KNN |
| Heuristics Method | GA | 2 | 80.41 | **82.99** | 80.93 | 78.87 |
| | BAT | 2 | 81.96 | **82.99** | 81.44 | 78.87 |
| | PSO | 3 | 79.90 | 81.96 | 82.47 | 75.77 |
| | FS | 3 | 80.41 | 80.41 | **82.99** | 74.23 |
| | AS | 2 | 80.41 | 82.47 | 82.47 | 75.26 |
| Exhaustive Method | GSS | 6 | 80.93 | 80.41 | 82.47 | 77.32 |
| | BF | 3 | 79.90 | 81.44 | 82.47 | 75.77 |
| | Average | | 80.56 | 81.81 | **82.18** | 76.58 |

**Table 12.** *Cont.*

| | | | | FILTER-BASED SUBSET SELECTION METHODS (FSS) | | |
|---|---|---|---|---|---|---|
| | | | ConsistencySubsetEval (CNS)Method | | | |
| **Attribute Evaluator** | **Search Methods** | **No. of Features** | **Performance Metrics (Accuracy)/Classifier** | | | |
| | | | **NB** | **DT** | **LR** | **KNN** |
| **Heuristics Method** | GA | 9 | 79.38 | 80.93 | **82.99** | 74.74 |
| | BAT | 17 | 78.87 | 80.41 | 81.44 | 77.84 |
| | PSO | 6 | 80.41 | 80.93 | 81.44 | 78.87 |
| | FS | 12 | 80.41 | 78.87 | 82.47 | 75.26 |
| | AS | 13 | 79.38 | 80.93 | 81.96 | 76.29 |
| **Exhaustive Method** | GSS | 6 | 79.90 | 80.41 | 81.96 | 78.87 |
| | BF | 5 | 80.93 | 80.93 | 81.96 | 77.32 |
| | Average | | 79.90 | 80.49 | **82.03** | 77.03 |
| | | | FILTER-BASED FEATURE RANKING METHODS (FFR) | | | |
| **Attribute Evaluator** | **Search Methods** | **No. of Features** | **Performance Metrics (Accuracy)/Classifier** | | | |
| | | | **NB** | **DT** | **LR** | **KNN** |
| **IG** | Ranker | 6 | 80.93 | **82.47** | 81.96 | 73.71 |
| **RFA** | Ranker | 6 | 78.87 | 79.90 | 80.93 | 67.53 |
| **GR** | Ranker | 6 | 80.41 | 81.96 | 81.96 | 74.23 |
| **CV** | Ranker | 6 | 79.38 | 78.87 | 80.41 | 74.74 |
| | Average | | 79.90 | 80.80 | **81.31** | 72.55 |

Boldface typeface indicates the highest value for each classifier.

**Table 13.** Performance Accuracy Values of FS-based Prediction Models on MW1 dataset.

| | | | | FILTER-BASED SUBSET SELECTION METHODS (FSS) | | |
|---|---|---|---|---|---|---|
| | | | CfsSubsetEval (CFS) Method | | | |
| **Attribute Evaluator** | **Search Methods** | **No. of Features** | **Performance Metrics (Accuracy)/Classifier** | | | |
| | | | **NB** | **DT** | **LR** | **KNN** |
| **Heuristics Method** | GA | 8 | 84.4 | 89.6 | 90 | 84 |
| | BAT | 9 | 86 | 90.4 | **91.2** | 84.4 |
| | PSO | 7 | 84.8 | 89.2 | 89.6 | 84 |
| | FS | 9 | 84.8 | 88.4 | 90 | 84 |
| | AS | 7 | 85.2 | 89.6 | 88.8 | 82 |
| **Exhaustive Method** | GSS | 6 | 84.4 | 89.2 | 90 | 84 |
| | BF | 7 | 84.8 | 89.6 | 89.2 | 82.8 |
| | Average | | 84.91 | 89.43 | **89.83** | 83.60 |
| | | | ConsistencySubsetEval (CNS)Method | | | |
| **Attribute Evaluator** | **Search Methods** | **No. of Features** | **Performance Metrics (Accuracy)/Classifier** | | | |
| | | | **NB** | **DT** | **LR** | **KNN** |
| **Heuristics Method** | GA | 11 | 85.2 | 89.2 | 90.8 | 83.6 |
| | BAT | 17 | 84 | 89.2 | 89.6 | 84.4 |
| | PSO | 8 | 85.6 | 88.4 | 90.4 | 82.8 |
| | FS | 17 | 83.6 | 89.6 | 90 | 82.8 |
| | AS | 13 | 83.6 | 89.2 | 90.4 | 83.6 |
| **Exhaustive Method** | GSS | 6 | 86 | **91.6** | 89.6 | 85.6 |
| | BF | 9 | 84.4 | 89.6 | 90.4 | 84.4 |
| | Average | | 84.63 | 89.54 | **90.17** | 83.89 |
| | | | FILTER-BASED FEATURE RANKING METHODS (FFR) | | | |
| **Attribute Evaluator** | **Search Methods** | **No. of Features** | **Performance Metrics (Accuracy)/Classifier** | | | |
| | | | **NB** | **DT** | **LR** | **KNN** |
| **IG** | Ranker | 6 | 83.6 | 90 | **90.4** | 84.8 |
| **RFA** | Ranker | 6 | 84.4 | 87.6 | 89.2 | 84 |
| **GR** | Ranker | 6 | 84.4 | **90.4** | 88.8 | 84.4 |
| **CV** | Ranker | 6 | 86.4 | 88.8 | **90.4** | 86.8 |
| | Average | | 84.70 | 89.20 | **89.70** | 85.00 |

Boldface typeface indicates the highest value for each classifier.

**Table 14.** Performance Accuracy Values of FS-based Prediction Models on PC2 dataset.

| FILTER-BASED SUBSET SELECTION METHODS (FSS) | | | | | | |
|---|---|---|---|---|---|---|
| **CfsSubsetEval (CFS) Method** | | | | | | |
| **Attribute Evaluator** | **Search Methods** | **No. of Features** | **Performance Metrics (Accuracy)/Classifier** | | | |
| | | | **NB** | **DT** | **LR** | **KNN** |
| **Heuristics Method** | GA | 5 | 94.18 | 97.51 | 97.51 | 96.54 |
| | BAT | 5 | 95.01 | **97.78** | 97.51 | 96.12 |
| | PSO | 5 | 94.46 | 97.51 | 97.65 | 96.81 |
| | FS | 5 | 93.91 | 97.65 | 97.51 | 95.84 |
| | AS | 6 | 95.57 | 97.51 | **97.78** | 95.29 |
| **Exhaustive Method** | GSS | 6 | 93.91 | 97.51 | 97.51 | 96.40 |
| | BF | 5 | 94.32 | 97.51 | 97.65 | 96.81 |
| | Average | | 94.48 | 97.57 | **97.59** | 96.26 |
| **ConsistencySubsetEval (CNS)Method** | | | | | | |
| **Attribute Evaluator** | **Search Methods** | **No. of Features** | **Performance Metrics (Accuracy)/Classifier** | | | |
| | | | **NB** | **DT** | **LR** | **KNN** |
| **Heuristics Method** | GA | 15 | 94.18 | 97.65 | 97.37 | 96.26 |
| | BAT | 17 | 94.46 | 97.51 | 97.51 | 96.12 |
| | PSO | 9 | 94.87 | 97.65 | 97.65 | 97.09 |
| | FS | 17 | 93.63 | 97.65 | 97.23 | 96.12 |
| | AS | 16 | 94.46 | 97.37 | 97.23 | 96.40 |
| **Exhaustive Method** | GSS | 6 | 93.91 | 97.65 | 97.65 | 95.84 |
| | BF | 1 | **97.78** | **97.78** | **97.78** | **97.78** |
| | Average | | 94.76 | **97.61** | 97.49 | 96.52 |
| **FILTER-BASED FEATURE RANKING METHODS (FFR)** | | | | | | |
| **Attribute Evaluator** | **Search Methods** | **No. of Features** | **Performance Metrics (Accuracy)/Classifier** | | | |
| | | | **NB** | **DT** | **LR** | **KNN** |
| **IG** | Ranker | 6 | 93.91 | 97.65 | 97.51 | 96.26 |
| **RFA** | Ranker | 6 | 96.68 | **97.78** | 97.65 | 96.12 |
| **GR** | Ranker | 6 | 94.18 | **97.78** | 97.09 | 96.12 |
| **CV** | Ranker | 6 | 95.84 | **97.78** | 97.51 | 95.98 |
| | Average | | 95.15 | **97.75** | 97.44 | 96.12 |

Boldface typeface indicates the highest value for each classifier.

From the aforementioned results, it is clear that there is no significant difference in the performance accuracy values of the FS methods. That is, the performance of FS methods depends largely on the dataset as the best subset of features that varied from one dataset to another. In addition, it was also observed that based on the FSS (CFS and CNS) methods, a varying number of features were selected. This presents a very interesting case on how the small number of features affects the performance of prediction models. Some studies argued that the lesser the features, the better the performance [26,37,43]. However, in this study, it was observed that the number of features selected depends largely on the FS method used. CNS often selects more features and the prediction models based on CNS outperforms other FS methods. In addition, as presented in Table 15, considering the FFR methods, IG had the best influence on the prediction models over other FFR methods. While considering the FSS, CNS based on BFS had the best influence on the prediction models. However, CFS based on BS had the best improvement on the performance of NB and DT and CNS based on GS and BF improved the performance of LR and KNN best, respectively.

**Table 15.** Performance (Accuracy) Variation of Prediction Models with different FS methods.

| FS Methods | Search Methods | NB | Variation (%) | DT | Variation (%) | LR | Variation (%) | KNN | Variation (%) |
|---|---|---|---|---|---|---|---|---|---|
| **No FS** | - | 81.14 | 0 | 84.50 | 0 | 85.93 | 0 | 80.54 | 0 |
| **FSS(CFS)** | GA | 83.39 | 2.77 | 86.25 | 2.06 | 86.04 | 0.12 | 81.67 | 1.40 |
| | BAT | **84.65** | **4.33** | **86.66** | **2.55** | 86.21 | 0.32 | 81.95 | 1.75 |
| | PSO | 83.24 | 2.58 | 86.17 | 1.97 | 86.61 | 0.79 | 81.52 | 1.21 |
| | FS | 83.18 | 2.52 | 85.52 | 1.21 | 86.44 | 0.60 | 81.10 | 0.70 |
| | AS | 84.56 | 4.22 | 86.13 | 1.92 | 86.09 | 0.19 | 80.57 | 0.03 |
| | GSS | 83.32 | 2.69 | 85.55 | 1.24 | 86.60 | 0.78 | 81.59 | 1.31 |
| | BF | 83.21 | 2.55 | 86.11 | 1.91 | 86.41 | 0.56 | 81.24 | 0.87 |
| **FSS(CNS)** | GA | 82.84 | 2.10 | 85.55 | 1.24 | **86.98** | **1.22** | 81.89 | 1.68 |
| | BAT | 82.49 | 1.67 | 85.61 | 1.31 | 85.93 | 0.00 | 82.26 | 2.14 |
| | PSO | 83.43 | 2.83 | 85.53 | 1.22 | 86.20 | 0.32 | 82.35 | 2.25 |
| | FS | 82.48 | 1.65 | 84.81 | 0.36 | 86.31 | 0.44 | 81.70 | 1.44 |
| | AS | 82.76 | 2.00 | 85.57 | 1.27 | 86.28 | 0.41 | 81.05 | 0.63 |
| | GSS | 83.37 | 2.75 | 86.47 | 2.33 | 86.42 | 0.57 | 81.57 | 1.28 |
| | BF | 84.35 | 3.96 | 85.92 | 1.68 | 86.76 | 0.97 | **83.70** | **3.92** |
| **FFR** | IG | 83.46 | 2.87 | 86.32 | 2.16 | 86.38 | 0.52 | 80.54 | 0.00 |
| | RFA | 82.54 | 1.73 | 85.55 | 1.24 | 86.27 | 0.40 | 80.68 | 0.17 |
| | GR | 83.27 | 2.58 | 86.28 | 2.08 | 86.16 | 0.27 | 80.67 | 0.17 |
| | CV | 83.64 | 3.03 | 85.38 | 1.03 | 86.16 | 0.27 | 81.09 | 0.69 |

Boldface typeface indicates the highest value for each classifier.

Furthermore, we conducted a stability test on the FS methods based on different prediction models using the average accuracy values from the experimental results. We calculated the Standard Deviation (SD) and the Co-efficient of Variation (CV) as presented in Appendix A Table A1. FFR methods produced more stable results in terms of accuracy across the prediction models as compared with the FSS methods. Consequently, even if there is no significant difference for the prediction models based on a variety of FS methods considered in this study, FFR proves to be more stable than FSS methods having lower CV values. The best prediction model developed using each classifier is illustrated in Figure 2. In Table 15, prediction models with FS methods outperform models developed without features selection. This indicates the importance of FS methods while developing the SDP model regardless of which family (characteristics) the classification algorithm belongs. Figure 3 shows the positive gain, that is, the variation of the prediction models with FS methods to prediction models without FS methods. Figures 4 and 5 show the performance stability of prediction models developed with FS methods respectively. Both figures depict the standard deviation (SD) and Co-efficient of Variation (CV) values for each FS method. Lastly, Figure 6 pictorially presents the performance FS method based on average accuracy values on each dataset.

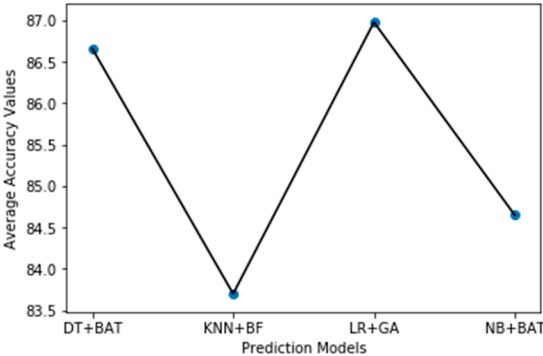

**Figure 2.** Average Accuracy for Prediction Models.

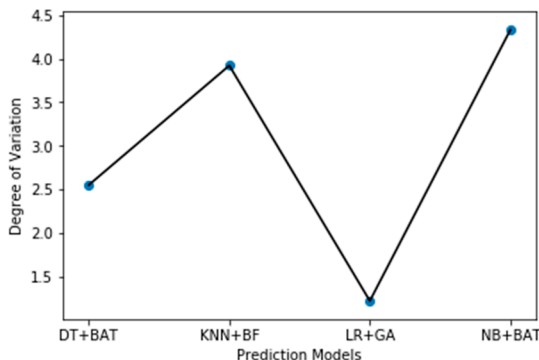

**Figure 3.** Prediction Models with the Highest Variation.

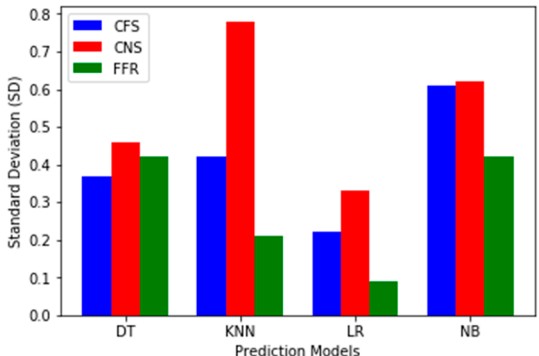

**Figure 4.** Performance Stability of Prediction Models (SD).

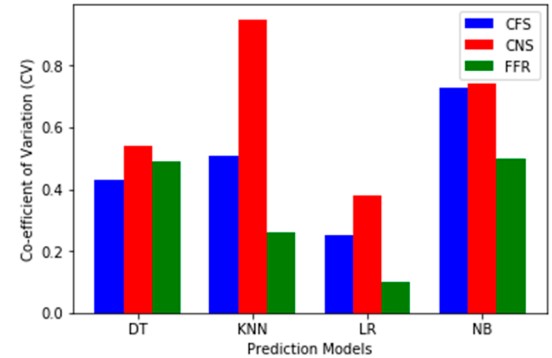

**Figure 5.** Performance Stability of Prediction Models (CV).

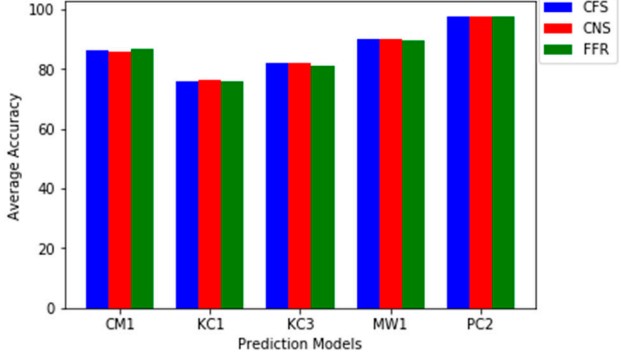

**Figure 6.** Performance of Feature Selection Method on each Dataset.

In conclusion, listed below are a summary of our findings from this comparative study:

- FS methods are very important and useful as it improves the performance of prediction models.

- Based on the individual performance accuracy values, FS methods had the highest improvement on the predictive performance of NB classifier.
- CFS based on (AS, BAT, GS, FS, PSO, BFS, and GSS) selects (automatically) the minimum number of features.
- On the average as presented in Table 15, CFS based on BAT had the highest positive variation on NB and DT while CNS based on GS and BFS had the highest positive variation value on LR and KNN.
- FFR had the lowest C.V. values which make it more stable than other FSS methods (See Table A1 in Appendix A).

## 5. Threat to Validity

This section discusses the threats to the validity of our comparative study. According to Wohlin et al. [44], empirical software engineering is becoming relevant and a vital factor of any empirical study is to analyze and mitigate threats to the validity of the experimental results.

*External validity*: This validity mainly bothers on the ability to generalize the experimental study. Five software defects datasets which have been extensively utilized in defect prediction were used in this study. Although these datasets differs in their characteristics (number of instances and attributes) and are from the commonly used corpora (NASA), we cannot generalize conclusions of this study on other software defect datasets. However, this study provided a comprehensive experimental setup, with applicable parameter tuning and settings, which makes it possible for researchers to replicate on other software defect datasets.

*Internal validity*: This validity stresses on the choice of prediction models and feature selection methods. Gao et al. [45] stated that factors such as choice of software applications, classification algorithm selection, and noisy datasets affect the internal validity of SDP. In this study, we selected 4 classificationn algorithms based on performance and heterogeneity (See: Table 4) and these classification algorithms are well used in SDP. Specifically, 18 methods based on two FS techniques with seven search methods were used in this study. Nonetheless, future studies can consider other FS techniques and new search methods.

*Construct validity*: This validity focuses on the choice of performance metrics used to evaluate the performance of prediction models. In this study, accuracy which measures the percentage of the correctly classified instances was employed and Co-efficient of Variation (C.V) was applied to the results of the prediction models to determine the performance stability of prediction models. However, other performance metrics such as Area under Curve (AUC) and F-Measure may also be applicable.

## 6. Conclusions and Future Work

SDP can assist software engineers in identifying defect-prone modules in a software system and consequently streamline the deployment of limited resources in Software Development Life Cycle (SDLC) during software development. However, the performance of SDP depends on the quality of software defect datasets which suffers from high dimensionality. Hence, the selection of relevant and irredundant features from software defect datasets is imperative to achieve a strong prediction model in SDP. This study conducted a comparative performance analysis via the investigation of eighteen FS methods on five software defect datasets from NASA repository with four classification algorithms. The FS methods were grouped into two Filter subset selections (FSS) (CFS and CNS) with seven different search methods (BFS, BAT, FS, AS, GSS, PSOS, and GS) and four Feature Filter Rank (FFR) (IG, RFA, GR, and CV) methods based on ranker search method. From the experimental results, IG recorded the best improvement on the prediction models over other FFR methods while CNS based on BFS had the best influence on the prediction models based on FSS methods. In addition, further analysis showed that prediction models based on FFR are more stable than other FS methods. It was conclusively discovered that the performance of FS methods varied across the dataset and that some classifiers behaved differently. This may be due to the class imbalance which is a primary data quality

problem in data science. In the future, we intend to look at how other data quality problems, such as class imbalance and outliers, affect FS methods in SDP.

**Author Contributions:** Conceptualization, A.O.B.; Investigation, A.O.B. and A.S.H.; Supervision, S.B.; Validation, S.J.A.; Writing—original draft, A.O.B.; Writing—review & editing, S.B., S.J.A. and A.S.H.

**Funding:** This research received no external funding.

**Acknowledgments:** This research was partly supported by Ministry of Higher Education Malaysia, under the Fundamental Research Grant Scheme (FRGS) with Ref. No. FRGS/1/2018/ICT04/UTP/02/04.

**Conflicts of Interest:** The authors declare no conflict of interest.

## Appendix A

**Table A1.** Performance Stability of FS methods on Prediction Models based on Average Accuracy Values.

| Classifiers | Metrics | CFS | CNS | FFR |
|:---:|:---:|:---:|:---:|:---:|
| NB | SD | 0.61 | 0.62 | **0.42** |
| | CV | 0.73 | 0.74 | **0.50** |
| DT | SD | **0.37** | 0.46 | 0.42 |
| | CV | **0.43** | 0.54 | 0.49 |
| LR | SD | 0.22 | 0.33 | **0.09** |
| | CV | 0.25 | 0.38 | **0.10** |
| KNN | SD | 0.42 | 0.78 | **0.21** |
| | CV | 0.51 | 0.95 | **0.26** |

Boldface typeface indicates the lowest value for each FS method.

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
