# Peer review of "Performance Analysis of Feature Selection Methods in Software Defect Prediction: A Search Method Approach"

_applsci, doi:10.3390/app9132764_

Round 1
Reviewer 1 Report
This paper can be accepted for publishing in Applied Sciences Journal, but some problem need to clarify as follow:
How many of the reduced datasets in Table 5 for this study?
…Deviation (SV) and … in line 337 is wrong, it should be (SD).
Why the performance of CNS is inferior to CFS in Figs 4 and 5?

Author Response
Point 1: 
 How many of the reduced datasets in Table 5 for this study?
Response 1: Thank you very much for this comment. Table 5 showed the datasets used in this study. The number of reduced datasets depends on the feature selection method used. However, all reduced features gotten from each feature selection methods were used in this study. Tables 10 – 14 has a column of a number of reduced features selected by each feature selection method in this study.
Point 2: …Deviation (SV) and … in line 337 is wrong, it should be (SD).
Response 2: Thank you very much for this comment. The correction has been made as highlighted.
… We calculated the Standard Deviation (SD) and the Co-efficient of Variation (CV)….
Point 3: Why the performance of CNS is inferior to CFS in Figs 4 and 5?
Response 3: Thank you very much for this comment. Figure 4 showed the performance stability of prediction models developed with feature selection methods based on standard deviation (SD). Figure 5 presented the performance stability of prediction models developed with feature selection methods based on the coefficient of variation (CV).
Figures 4 and 5 are based on different evaluation metrics (SD and CV respectively). In addition, both CfsSubsetEval (CFS) and ConsistencySubsetEval (CNS) methods belong to Filter-based Subset Selection (FSS) Methods. On average, prediction models based on CFS have lower SD and CV values in comparison with prediction models based on CNS. That is, the CFS based prediction models are more stable (low SD and CV values) than CNS based prediction models.

Reviewer 2 Report
In order to highlight conclusions, results and findings from the experimental study , I believe that the analysis should include a feedback on the related works. I mean, results of this performance study should be related and compared with those available in the literature cited in section 2. In this way, such performance study could be better justified by all interested readers.
Author Response
Point 1: In order to highlight conclusions, results, and findings from the experimental study, I believe that the analysis should include feedback on the related works. I mean, results of this performance study should be related and compared with those available in the literature cited in section 2. In this way, such a performance study could be better justified by all interested readers.

Response 1: Please provide your response for Point 1. (in red)
Thank you very much for this comment. Related studies cited in Section 2 has been carefully related to the experimental analysis as indicated by the reviewer. Specifically, line 287 – 292
…. Our findings on the positive effect of FS methods on prediction models are in accordance with research outcomes from existing empirical studies. Ghotra, et al. [28], Afzal and Torkar [26], Akintola, et al. [42] in their respective studies also reported that FS methods had a positive effect on prediction models in SDP. However, our study explored the effect of FS methods on prediction models based on the search method which is different from existing empirical studies…
And line 321 – 323.
This research outcome is related to the findings from Xu, et al. [29], Kondo, et al. [31] and Muthukumaran, et al. [30]. Although on average, FSS methods proved to be better than FFR methods.

Reviewer 3 Report
This performance evaluation study focuses on Software Defect Prediction (SDP) models. The quality of SDP models depends mostly on the quality of software repositories (dataset) used in building the SDP models. High dimensionality is one of the data quality problems that affect the performance of SDP models. The choice of Feature Selection (FS) methods in SDP is still a problem as most empirical studies of FS methods in SDP produce contradictory and inconsistent results. This paper evaluates several filter feature ranking (FFR) and filter subset selection (FSS) methods. The results showed that prediction models based on FFR proved to be more stable than the prediction models based on FSS methods.
To be honest, I was very sceptic after I have read the abstract. Many software defection prediction papers pretend to have invented a kind of a "crystal ball". This paper is different to some degree. I was very positively surprised because the article explains quite well the complexities and interdependencies between feature ranking, feature selection and resulting classification problems. That all have to be considered and that depends on the analyzed code base (dataset) which makes it hard to develop proper software defect prediction models.
The paper is well written and understandable, and it fits the intended scope of the artificial intelligence section of the journal.
I have only (non-severe) minor revision recommendations to improve the overall readability of the paper and to improve the presentation of the results.
1. The study should reason what has been the reason that precisely the mentioned FFR, CFS, and CNS approaches have been selected for analysis. This selection drops somehow from heaven. There are more approaches around. So, what were the inclusion/exclusion criteria for this study?
2. The interplay and interdependencies between FFR, FSS and classification should be explained at the beginning of Section 3. Subsection 3.1 would be better understood if Subsection 3.4 (Experimental Setup) had been introduced before. So, I recommend shifting Subsection 3.4 (Experimental Setup) before Subsection 3.1.
3. What is more, there are several sections having the 3.4 number. The authors should correct their subsection numbering.
4. Table 2 and Table 3 seem to be very redundant and can be combined.
5. This is my main point: The study works with five software defect datasets including software programmed in C, C++, and Java. The question arises whether the results might contain some biases because of some specifics of these three languages/software types. I recommend reflecting on these aspects in a systematic and additional Threat on Validity section. This section should reason about internal and external threats on validity and what can be concluded from this study and what should not be concluded from this study. It would be a good academic practice to reason on the limitations of the presented approach.
6. The authors should create and provide a public git repository containing their prediction models that have been built using the WEKA machine learning framework. It would be good academic practice as well.
7. This comment applies to Tables 6-9 and Tables 10-14. The detailed tables should be provided in an Appendix, and within the text, only the best performers should be displayed. These long tables disturb the "reading flow" of the paper significantly. But this can be healed easily shifting all the detail data into an appendix.
8. Figure 4 and 5 are redundant with Table 16. So, Table 16 could be even deleted (or shifted into an appendix).
9. Furthermore, I recommend for Figures 2 - 5 to use a more "academic" looking charting tool suite like matplotlib. What is more, I wonder whether it is possible to display/compare all analyzed combinations graphically. It would be beneficial for the reader to understand the differences (advantages/shortcomings) better.

Author Response
Point 1: The study should reason what has been the reason that precisely the mentioned FFR, CFS, and CNS approaches have been selected for analysis. This selection drops somehow from heaven. There are more approaches around. So, what were the inclusion/exclusion criteria for this study?

Response 1: Thank you very much for this comment. From existing empirical studies, supervised feature selection (FS) methods are the type of FS methods used in software defect prediction (SDP). Filter Feature Ranking (FFR) and Filter Feature Selection (FSS) are the prominent types of supervised FS methods used in SDP. In addition, other forms of FS methods such as Wrapper FS methods and Embedded FS methods are heavily dependent on the choice of prediction models which leads to biases in the outcome. The aforementioned reasons are the basis for the selection of FFR and FSS (CNS and CFS) methods in this study.
See line 72-79:
… Supervised feature selection techniques evaluate the available feature’s characteristics and derive a set of pertinent characteristics based on labeled datasets. The criteria used to prove the useful characteristics depend on the nature of the technique utilized. Filter feature-ranking (FFR) methods which are types of supervised FS methods depend on each feature by certain critical factors, and then the analyst culls some features that are congruous with a particular dataset. On the other hand, filter subset selection (FSS) methods which are another type of supervised FS methods, search for a subset of features that have good predictive capability collectively...
Point 2: The interplay and interdependencies between FFR, FSS, and classification should be explained at the beginning of Section 3. Subsection 3.1 would be better understood if Subsection 3.7 (Experimental Setup) had been introduced before. So, I recommend shifting Subsection 3.7 (Experimental Setup) before Subsection 3.1.

Response 2: Thank you very much for this comment.
Firstly, there are no interdependencies or interplay between the FSS, FFR and the classification methods. Each concept has been outlined as a subsection and detailed description has been given in accordance with this study.
Secondly, Subsection 3.6 (Experimental Setup) has been shifted but to Subsection 3.4. We agree with the recommendation of the reviewer that the experimental setup should be moved. However, in the description of the experimental setup in subsection 3.4 (See: line 204 – 230), we referenced the Tables in Subsection 3.1 to 3.3. This makes it hard to place subsection 3.6 (experimental setup) to subsection 3.1 as it will distort the flow.
Point 3: What is more, there are several sections having the 3.4 number. The authors should correct their subsection numbering.

Response 3: Thank you very much for this comment. The subsection numbering has been corrected as mentioned.
Point 4: Table 2 and Table 3 seems to be very redundant and can be combined.

Response 4: Thank you very much for this comment. We deliberately split the information into two separate tables (Table 2 and Table 3). Table 2 gives a description of the FSS methods with the search methods used in this study. However, Table 3 presented the parameter settings of the search methods used with each of the FSS (CNS and CFS) methods.
Combining both tables will make it less interesting as there will be much information in a single table.
Point 5: This is my main point: The study works with five software defect datasets including software programmed in C, C++, and Java. The question arises whether the results might contain some biases because of some specifics of these three languages/software types. I recommend reflecting on these aspects in a systematic and additional Threat on Validity section. This section should reason about internal and external threats on validity and what can be concluded from this study and what should not be concluded from this study. It would be a good academic practice to reason on the limitations of the presented approach.
Response 5: Thank you very much for this comment. A new section which is on Threat to Validity has been added to the main text to analyze and mitigate the threats to the validity of the experimental results. See line 386 – 409.
Point 6: The authors should create and provide a public git repository containing their prediction models that have been built using the WEKA machine learning framework. It would be good academic practice as well.
Response 6: Thank you very much for this comment. The git repository containing the prediction models and datasets used in this study has been cited accordingly. See line 255 – 256.
… For reproducibility, all models and datasets used in this study are readily available in [42] …
Point 7: This comment applies to Tables 6-9 and Tables 10-14. The detailed tables should be provided in an Appendix, and within the text, only the best performers should be displayed. These long tables disturb the "reading flow" of the paper significantly. But this can be healed easily shifting all the detail data into an appendix.
Response 7: Thank you very much for this comment. After careful consideration, we believe shifting Tables 6 – 14 to the appendix will distort the flow of the manuscript based on the following reason.
1. The results displayed in the respective tables are referenced and discussed in the main text.
2. Tables 6-9 showed the performance (based on accuracy) of prediction models with and without FS methods. This is to show the effect of FS methods on prediction models.
3. Tables 10-14 presented the detailed effects of each FS methods on each dataset. This is to deduce how an FS method behaves with different datasets.
However, we shifted Table 16 to appendix based on the recommendation of the reviewer.
Point 8: Figure 4 and 5 are redundant with Table 16. So, Table 16 could be even deleted (or shifted into an appendix).
Response 8: Thank you very much for this comment. As recommended, Table 16 has been moved to appendix and cites accordingly in the main text.
Point 9: Furthermore, I recommend for Figures 2 - 5 to use a more "academic" looking charting tool suite like matplotlib. What is more, I wonder whether it is possible to display/compare all analyzed combinations graphically. It would be beneficial for the reader to understand the differences (advantages/shortcomings) better.
Response 9: Thank you very much for this comment. Figures 2-5 has been redrawn using matplotlib as recommended.
